# Nanobodies as Diagnostic and Therapeutic Tools for Cardiovascular Diseases (CVDs)

**DOI:** 10.3390/ph16060863

**Published:** 2023-06-09

**Authors:** Lorena-Andreea Bocancia-Mateescu, Dana Stan, Andreea-Cristina Mirica, Miruna Gabriela Ghita, Diana Stan, Lavinia Liliana Ruta

**Affiliations:** 1DDS Diagnostic, 7 Vulcan Judetu, 031427 Bucharest, Romania; research@ddsdiagnostic.com (L.-A.B.-M.); dana_stan@ddsdiagnostic.com (D.S.); i.andreeacristina@yahoo.com (A.-C.M.); assistant.research3@ddsdiagnostic.com (M.G.G.); dianastan335@gmail.com (D.S.); 2Medicine Doctoral School, Titu Maiorescu University, 031593 Bucharest, Romania; 3Advanced Polymer Materials Group, University Politehnica of Bucharest, 1–7 Gh. Polizu Street, 011061 Bucharest, Romania; 4Faculty of Chemistry, University of Bucharest, 90-92 Panduri Street, 050663 Bucharest, Romania

**Keywords:** theragnostic, nanobody, imaging

## Abstract

The aim of this review is to summarize some of the most recent work in the field of cardiovascular disease (CVD) diagnosis and therapy, focusing mainly on the role of nanobodies in the development of non-invasive imaging methods, diagnostic devices, and advanced biotechnological therapy tools. In the context of the increased number of people suffering from CVDs due to a variety of factors such as sedentariness, poor nutrition, stress, and smoking, there is an urgent need for new and improved diagnostic and therapeutic methods. Nanobodies can be easily produced in prokaryotes, lower eukaryotes, and plant and mammalian cells, and offer great advantages. In the diagnosis domain, they are mainly used as labeled probes that bind to certain surface receptors or other target molecules and give important information on the severity and extent of atherosclerotic lesions, using imaging methods such as contrast-enhanced ultrasound molecular imaging (CEUMI), positron emission tomography (PET), single-photon emission computed tomography coupled with computed tomography (SPECT/CT), and PET/CT. As therapy tools, nanobodies have been used either for transporting drug-loaded vesicles to specific targets or as inhibitors for certain enzymes and receptors, demonstrated to be involved in various CVDs.

## 1. Introduction

The American Heart Association together with the National Institutes of Health publishes a report every year with the newest data on heart disease, stroke, and cardiovascular risk factors [1]. In the United States, cardiovascular disease continues to be the leading cause of death [2]. Many common habits, along with increasingly hectic lifestyles, have become the main triggers of CVDs, along with other significant risk factors, including age, smoking (the majority of adolescents using flavored electronic cigarettes), obesity, hypertension, dyslipidemia, and diabetes mellitus [3]. From 2013 to 2020, the overall CVH (cardiovascular health) score was 73.6 for all teenagers between 16 and 19 years in the USA and 62.5 for all adults [4].

In European countries, cardiovascular diseases represent one of the leading causes of death, with around 23.8% of patients suffering from CVDs in France, 21.6% in Denmark, 50 to 60% in Romania, and 65.4% in Bulgaria [5].

Nutrition plays a key role in reducing the risk of cardiovascular disease, with overweight and obesity being increasingly common, especially among young people and adults. An extensive study conducted in 2021 ascertained that every 5 kg/m^2^ increase in body mass index was correlated with a 15% increased risk for congenital heart defects (CHD), 23% increased risk for atrial fibrillation (AF), 41% increased risk for heart failure (HF), and 49% increased risk for hypertension. Additionally, one of the leading causes of morbidity and mortality in the world is coronary artery disease (CAD), which can be clinically described as anything from stable angina to acute coronary syndromes [6]. Hence, all of these factors combined led to some worrying statistics; globally, in 2020, 19.05 million CVD-related deaths were predicted, with an increase of 18.71% since 2010.

Progresses in the diagnosis and treatment of CVDs have been made, forced by the increase in the number of patients and the spread to young people and children. Congenital heart diseases should not be neglected, since almost 500,000 adults in the USA suffer from congenital heart disease [7].

The diagnostic methods used today are routine, complex methods that require equipment and technologies that are not always readily available and require overqualified personnel. The diagnosis of CVDs is performed using the following methods: electrocardiogram (ECG), echocardiogram, chest X-ray, pulse oximetry, and cardiac catheterization. The open-heart surgery and catheter method are usually used for treating the majority of cardiovascular diseases [8]. Data mining is also a cutting-edge tool used in totally different fields, with applications on the medical side, especially in heart disease diagnosis [9].

Just as Einthoven’s discovery and development of the electrocardiogram was a real breakthrough for the medical field [10], the discovery of nanobodies brings with it huge potential for the diagnosis and treatment of many diseases. Nanobodies or single-domain antibody fragments are found in sharks and camelids [11]. Despite their low molecular weight—they are approximately one-tenth the size of more common antibodies—they selectively bind to antigens with high affinity [12]. Compared to full-size antibodies, which may circulate for days, nanobodies have substantially faster pharmacokinetics due to their low molecular weight.

These considerations have particular importance when molecular targets localize in proximity to the blood pool because long clearance times would yield unfavorable target-to-background ratios for antibodies. Nanobodies retain the specificity and outstanding affinity in the low nanomolar range characteristic of antibodies. Some other medical researchers have used nanobodies as affinity ligands [13].

The current paper brings into light the role of nanobodies in some of the most recent advances in the cardiovascular disease domain, everything from screening and prevention methods to diagnostic and therapies. The main purpose is to expose both the strengths and shortcomings of the newly developed diagnostic and therapy methods and to help further the development of the field, based on some very promising results, that have led to new approved medicines and non-invasive diagnostic methods. The paper answers questions related to nanobody production and challenges, the efficacy of nanobody-based screening and diagnostic methods, and their role as guiding molecules for drug-loaded vesicles and inhibitors. Although there are a series of studies regarding the use of VHH as imaging and therapeutic agents, we selected some of the most recent work in the field, with respect to our proposed topic, which is focused on CVDs. Thus, the article comprises four main chapters:Antibodies versus nanobodies: sums up the most significant differences between antibodies and nanobodies and the specific shortcomings;How to produce nanobodies: dedicated to the methods for obtaining nanobodies, especially in yeasts;Nanobodies for CVD diagnosis: the most recent and promising discoveries for the diagnostic of CVDs;Nanobodies for cardiovascular disease therapy: illustrates the therapeutic potential of nanobodies and reviews the latest achievements in the field.

Although nanobodies have received a lot of attention in the last few years and there is a plethora of literature on this topic, when it comes to the cardiovascular disease domain, there is no recent available review article focused on prevention, diagnosis, and therapy.

## 2. Antibodies Versus Nanobodies

Antibodies are ubiquitous reagents in research and diagnostics, allowing the detection of many molecules [12].

The use of animals, especially mammals, to generate IgG antibodies is expensive, raises ethical issues, and requires considerable effort. Mouse and rabbit antibodies are tools for many basic techniques and medical diagnostic tests. These primary antibodies are often detected or immobilized indirectly by polyclonal anti-IgG secondary antibodies. The need for a continuous supply of anti-IgG sera requires the storage, immunization, bleeding, and ultimately killing of large numbers of goats, sheep, rabbits, and donkeys [14]. In addition, each new batch of serum contains another heterogeneous mixture of antibodies, which must be purified and then isolated (by depletion) from non-specific antibodies and cross-reactions. In addition, the success of this procedure must be quality controlled each time. The large size of secondary antibodies is also a disadvantage, as it limits tissue penetration and introduces considerable displacement of markers, which reduces the image relocation that can be obtained by super-resolution fluorescence microscopy methods [15,16,17]. Their non-combining nature further precludes genetic engineering (tagging or fusion with reporter enzymes) (Figure 1).

Therefore, the discovery of heavy chain-only antibodies from camelids [19] or nanobodies has a significant impact on antibody technology and production.

Camelid-derived single-domain antibodies (SdAbs) or the variable domains of llama heavy-chain antibodies (VHH) or nanobodies (Nbs) are the smallest antibody fragments with a full antigen-binding capacity [20]. They were first discovered in camelids in 1993 by Hamers-Casterman and his collaborators, which revealed that they contain CH2 and CH3 constant domains, a hinge region, and a variable heavy-chain domain (VHH) [19,21].

Nanobodies consist of about 120 amino acids with a molecular weight of 12–15 kDa [22] and a size of 4 nm in length and 2.5 nm in diameter [23]. They penetrate safely into intercellular spaces and tissues due to their small molecular size.

Even discovered nearly 30 years ago, nanobodies are already widely used by the biotechnology research community [19]. In addition, a number of nanobodies are under clinical investigation for a broad spectrum of diseases, including inflammation, breast cancer, brain tumors, lung diseases, cardiovascular diseases, and infectious diseases [15,20,24]. The easy fabrication of nanobodies and their superior properties (small size, high stability, strong antigen binding affinity, water solubility, and natural origin and reversible refolding) [25,26] make them a useful tool to develop sensitive, selective, and effective therapeutic agents.

## 3. How to Produce Nanobodies: The Main Address on Yeast

The development of nanobodies by animal immunization is one of the most used approaches, leading to selective antibody clones owing to the powerful evolutionary process of somatic hypermutation. However, this method is time-consuming, slow, not always accessible, and poorly compatible with many antigens [27].

The pharmaceutical industry faces many challenges regarding the production of antibodies due to high production costs, processing time, and low production of antibodies in mammalian cells. Around 89% of approved biopharmaceuticals are obtained using molecules produced in mammalian cells and microorganisms. Nanobodies are relatively easy to produce in prokaryotes, lower eukaryotes, and even plant and mammalian cells, and some molecules have poor secretion activity. One of the common choices for nanobody production is using the bacterial expression system, such as Escherichia coli, due to its easy purification and low cost [28]. At the same time, the model organism Saccharomyces cerevisiae is a well-characterized cell factory, capable of post-translational modifications and protein secretion, which facilitates subsequent isolation and purification [29].

In yeast, the heterologous proteins are usually produced by targeting the secretory pathway, which enables the secretion of soluble, functional, correctly folded, and N-glycosylated proteins to the media [30].

For the proper folding of VHH in vivo, Gorlani et al. (2012) identified five key residues critical for folding and secretion. All key residues are localized in the V segment, in the proximity of the J segment of VHH, and this type of localization could influence the production yield. These residues are amino acids L20, W36, R38, D86, and Y90, which behave like Vendruscolo’s key residues. They speculated a possible two-step folding mechanism for VHH and the N-terminal V segment folds first, proved by the fact that the key residues make a cavity where the J segment subsequently docks [31].

Yeast cell surface display (YSD) is a technology for engineering proteins. Using this method, a protein of interest is genetically fused to a cell surface anchor protein and covalently tethered to the cell wall via a glycosylphosphatidylinositol (GPI) attachment signal. The activities of the displayed proteins can be quickly and quantitatively evaluated using flow cytometry. This technique has great advantages in eukaryotic cells, due to the folding machinery and safe pathogen-free protein production [32]. YSD followed by cell sorting is commonly used in nanobody recovery. The major advantage of this method is the compatibility with flow cytometry for quantitative and multi-parameter analysis [33].

A new full platform for nanobody discovery, based on an engineered library displayed on Saccharomyces cerevisiae using YSD, was reported. The library generates nanobody binders available for crystallization with its antigen and amenable to subsequent structural determination. The platform discovered conformationally-selective nanobodies to two distinct human G protein-coupled receptors (GPCR), a family of transmembrane receptors in humans with a role in cardiovascular biology, revealing a potential tool for structural studies [34].

In the same line, Kajiwara et al. (2021) utilized the SpyTag/SpyCatcher system to ligate the nanobodies and anchor proteins. Additionally, they designed two gene cassettes to fuse a synthetic anchor protein with a SpyCatcher and a nanobody with SpyTag. The final product is transported via the secretion pathway and anchored to the cell wall via a GPI attachment signal. They obtained 90% functional nanobodies on yeast cell surfaces and no intercellular protein ligation events [32].

Despite its major advantages, as a cell factory, S. cerevisiae also has some limitations as an antibody expression host due to its native hyper mannose glycosylation, which could be highly antigenic for humans. This problem overlaps with the fact that many antibody fragments do not have glycosylation sites [29]. Wang et al. (2021) reported the production of three non-glycosylated antibody fragments from humans and the Camelidae family in different S. cerevisiae lines, with a high capacity for α-amylase secretion. These antibody fragments include a nanobody, consisting of a single V-type domain (Nan). To demonstrate the binding specificity of the nanobody fragment to the target protein lysozyme, a molecule wearing a C-6xHis-tag was selected [29].

In order to improve VHH expression levels, both episomal and integrating DNA vectors were developed. Nanobodies obtained through this procedure are suitable for any application that requires a double antibody sandwich (ELISA, rapid immunochromatographic assays) [35].

A recent paper described an engineered technology for synthetic recombinant antibody generation, mimicking the somatic hypermutation, based on orthogonal DNA replication (OrToREp) and YSD, called autonomous hypermutation yeast surface display (AHEAD). This system (AHEAD) obtained potent nanobodies against the SARS-CoV-2 S glycoprotein, GPCR, or other targets, emphasizing the significant advantage of using yeast for developing nanobodies, as opposed to animal immunization [27].

The Aga2p subunit is used in S. cerevisiae cells to display up to a hundred thousand copies of an affinity reagent (e.g., nanobody) fused at its C-terminal to the N-terminal end. Uchanski et al. (2019) revealed a new pNACP vector system designed to display the fusion proteins containing nanobodies followed by Aga2 and ACP (acyl carrier protein). The system exhibited a high displayed level of cloned nanobodies against the human α2A adrenergic receptor, the human OX2 orexin receptor, and human coagulation Factor IX and it was monitored using a covalent fluorophore attached to the ACP tag. [33].

Another yeast used for nanobody production is Pichia pastoris. In the yeast Pichia spp., the production is driven under the expression vectors developed for this system based on the control of the alcohol oxidase 1 promoter (AOX1). Due to the multiple disadvantages of the methanol-induced system, the constitutive GAL (glyceraldehyde-3-phosphate dehydrogenase) promoter could be a good alternative for AOX1, because the expression level of heterologous proteins is compatible with the AOX1 system. Chen and collaborators reported the efficient expression of the anti-CEACAM5 nanobody 11C12 gene under the control of the GAP promoter Carcinoembryonic antigen (CEA, an important tumoral marker) [36].

## 4. Nanobodies for CVD Diagnosis

### 4.1. Atherosclerosis

Atherosclerosis is the main cause of cardiovascular diseases, including heart failure, myocardial infarction, and stroke [37].

To generate a non-invasive method for evaluating the risk degree for patients with atherosclerotic disease, to suffer a stroke or myocardial infarction, scientists used nanobodies coupled with lipid-shelled decaflurobutane microbubbles to evaluate VCAM-1 expression, using contrast-enhanced ultrasound molecular imaging (CEUMI). This technique was tested in both a murine model and on human tissue and the results show that it could be used to detect vascular inflammation in the early stages of atherosclerosis. Lipid-coated decafluorobutane microbubbles were functionalized with maleimide to be covalently conjugated with the modified nanobodies bearing a C-terminal cysteine thiol group. The nanobodies were labeled with N-hydroxysuccinimide-fluorescein to quantify the yield of conjugation with the microbubbles. CEUMI results demonstrating VCAM-1 expression in mouse aorta showed an approximately threefold increase in microbubbles with cAbVcam1-5 compared with control microbubbles conjugated with nanobodies (MBVHH2E7) in double knockout (DKO) mice with early-stage atherosclerosis at 10 weeks of age and a twofold increase at 40 weeks of age. However, some potential flaws of this study have been identified, such as low efficiency of the bioconjugation, the animal model was not the optimum choice, and that no normal arterial tissue was used in this study, due to ethical reasons [38].

Another imaging technique using nanobody probes is positron emission tomography (PET). This was used to measure the metabolic activity of cells affected by atherosclerosis using nanobodies targeting various receptors such as macrophage mannose receptor (MMR), lectin-like oxidized LDL receptor (LOX), and VCAM-1 [39]. Translational PET/magnetic resonance imaging was used in rabbit models of atherosclerosis. For this purpose, nanobodies were labeled with various radiotracers, of which the ^68^Ga-MMR nanobody was particularly prominent because its uptake was correlated with macrophage accumulation during the progression of atherosclerosis. Therefore, these techniques using nanobodies could be used in personalized medicine and treatment monitoring.

Targeting atherosclerosis using anti-MMR nanobodies was investigated by Bala et al. (2018), but the results did not meet expectations. They investigated in vivo nuclear imaging SPECT/CT of technetium-99m (Tc-99m)-labeled nanobody MMR3.49 in apolipoprotein E-deficient (ApoE^−/−^) mice in parallel with autoradiography and immunofluorescence. Although good biodistribution of the Tc-99m-labeled anti-MMR nanobody was observed in the liver, spleen, heart, thymus, bone marrow, and lymph nodes, the in vivo studies did not proceed as desired, presumably due to the mouse model used for the testing. By using SPECT/CT and PET/CT imaging methods, Bala et al. (2018) assessed the ability of the nanobody cAbMMR3.49 to target the macrophage mannose receptor (MMR), which is expressed by macrophages in atherosclerosis. Anti-MMR nanobodies were functionalized with 99mTc for SPECT/CT imaging and administered to Apolipoprotein E-deficient (ApoE^−/−^) mice, whereas ^18^Fluor was used to label the nanobodies for PET/CT imaging on Watanabe Heritable Hyperlipidemic (WHHL) rabbits. Both ApoE^−/−^ and control mouse MMR-positive tissues, including the liver, spleen, and lymph nodes, showed high absorption of the 99mTc-cAbMMR3.49 nanobody. Although ApoE^−/−^ was highly absorbed in the atherosclerotic aorta, the concentration of nanobodies was lower in atherosclerosis with a higher score. PET/CT imaging showed that the nanobody accumulated not only in the aorta of the rabbit model of atherosclerosis but also in control rabbits. Although the nanobody cAbMMR3.49 binds specifically to the MMR receptor, attempts to image macrophage mannose receptors with it have been unsuccessful [40].

An anti-VCAM-1 nanobody, called cAbVCAM-1-5, was radiolabeled with Fluorine-18 (^18^F) in order to analyze the in vivo behavior of atherosclerotic lesions in apolipoprotein-deficient (ApoE^−/−^) mice. Using a necrosing factor, the [^18^F]FB-anti-VCAM-1 nanobody’s functionality and selectivity were examined in vitro on the mouse endothelial cell line bEND5. According to the gamma-counter findings, 5 nM of [^18^F]FB-cAbVCAM-1-5 only binds to VCAM-1 positive cell. In comparison to the control nanobody, the uptake of the nanobody in the excised aorta segments of mice was considerably greater at score 1 (1.18 + 0.36%ID/g), score 2 (1.49 + 0.37%ID/g), and score 3 (1.61 + 0.41%ID/g). According to PET/CT images, the labeled nanobody accumulated at the ascending aorta and aortic arch level, primarily in the segments with Score 3. As a consequence, the atherosclerotic lesions present in that region were particularly recognized using the [^18^F] FB-cAbVCAM-1-5 nanobody [41].

By chemically modifying a nanobody and covalently fusing it to azidified silicon wafers, Biacore^TM^ C1 sensor chips, and boron-doped microcrystalline diamond films, Duy Tien Ta (2016) created a biosensor platform that can detect the VCAM-1 atherosclerosis biomarker. Using the copper(I)-catalyzed Huisgen 1,3-dipolar azide-alkyne cycloaddition (CuAAC) process, the nanobodies employed in this work were functionalized with an alkyne function at the C-terminus to be conjugated to an azide-functionalized substrate. The developed platforms enhanced the nanobodies’ sensitivity and ability to attach to the antigen, and the CuAAC approach helped to align their surfaces with the substrate [42].

Radiolabeled nanobodies were used for in vitro diagnosis by flow cytometry analysis of mouse bEND5 endothelial cells and human HUVEC endothelial cells stained with an anti-VCAM1 antibody. This showed that 6 of 10 nanobodies exhibited cross-reactivity with the hVCAM1 protein at nanomolar levels. cAbVCAM1-5 was selected for in vivo analysis because it exhibited the highest heat resistance and production yield. In vivo imaging of atherosclerotic lesions was performed by injecting VCAM1-targeted nanobodies radiolabeled with Tc-99m (^99m^Tc-cAbVCAM1-5) into ApoE-deficient (ApoE^−/−^) mice. SPECT/CT imaging showed that the nanobodies were beneficial for the detection of vascular cell adhesion molecule 1 and had the potential for rapid clinical translation [43].

The progression of atherosclerosis in mice and rabbits was evaluated using radiolabeled nanobodies with ^64^Cu for the screening of specific biomarkers such as (VCAM)-1, lectin-like oxidized low-density lipoprotein receptor-1 (LOX-1), and MMR. The nanobodies functionalized with 1,4,7-triazacyclononane-1,4,7-triacetic acid showed good results for in vivo screening in ApoE^−/−^ mice, with ^64^Cu deposition observed at the aortic root for the MMR nanobody in particular. In addition, the MMR nanobody was labeled with gallium-68 (^68^Ga) to phenotype rabbit atherosclerotic aortas using PET/MRI imaging compared with clinically available radiotracers ^18^F-FDG and ^18^F-NaF. In rabbits with advanced atherosclerotic lesions, microcalcification, macrophage burden, and vessel wall inflammation were observed with PET/MRI, demonstrating that aortic uptake was high for ^18^F-FDG, ^18^F-NaF, and ^68^Ga-MMR [44].

### 4.2. Coronary Heart Disease (CHD)

In several signal transduction pathways necessary for angiogenesis and cell migration, Vascular Endothelial Growth Factor Receptor -2 (VEGFR-2) plays a crucial role. Behdani et al. (2012) investigated the binding efficiency of several nanobodies to the VEGFR-2 factor using surface plasmon resonance (SPR) analysis. All nanobodies tested recognize the same epitope, but two nanobodies exhibited higher affinity, namely 3VGR19 and 4VR38 with KD values between 5.4 and 6.8 nM [45].

The in vitro and in vivo specificity and selectivity of nanobodies for the detection of VEGFR-2 in tumors were investigated. The in vitro binding affinity of the 8B6 nanobody was demonstrated by FACS analysis through the interaction of the nanobody with EGFR-overexpressing cells such as A431, HER14, and DU145. For the in vivo analysis, the nanobody was labeled with ^99m^Tc and injected intravenously into six healthy −/− mice. Radioimmune-imaging results SPECT show that the nanobodies were rapidly eliminated from the blood due to their low molecular weight and distributed mainly in the renal system and to a lesser extent in the liver [46].

Nanobodies against mouse vascular endothelial growth factor (mVEGF) were tested for specificity in multiple ELISA assays and phage displays. There was no cross-reactivity between the nanobodies (Nb5 and Nb10) with bovine serum albumin (BSA), casein, human VEGF, epidermal growth factor, and basic fibroblast growth factor. The nanobodies will be used for diagnostic and therapeutic purposes in future studies [47].

An immunosensor for the detection of the coronary heart disease marker apolipoprotein B-100 (ApoB-100) was developed using a glass carbon electrode, functionalized with streptavidin, and 25 μL of biotinylated anti-ApoB-100 nanobody (BiNb4) at a concentration of 100 μg/mL. Various concentrations of ApoB-100 were measured by electrochemical impedance spectroscopy (EIS), and the detection limit was 0.03 ng/mL. Investigations were conducted on the immunosensor’s specificity by EIS measurements on 1 ng/mL BiNb4 in parallel with 10 ng mL-1 human chorionic gonadotropin (HCG), carcinoembryonic antigen (CEA), prostate-specific antigen (PSA), and alpha-fetoprotein (AFP). No cross-reactivity was observed. [48].

Patients with cardiovascular disorders can be clinically tested for apolipoprotein-A1 (Apo-A1) using an ELISA immunoassay. An immunosensor based on a similar principle was developed, using a screen-printed carbon electrode (SPCE) enhanced with gold nanoparticles. After the binding affinity studies, two nanobodies were taken into consideration, Nb 11 as a capture antibody and a silver nanoparticle-loaded nano-hydroxyapatite (Ag-nHAP)-modified Nb 19 as detection. Linear sweep stripping voltammetry (LSSV) analysis was used to generate the calibration curve for the detection of Apo-A1 at concentrations between 10^−5^ and 50 ng/mL. The detection limit was 0.02 pg/mL, resulting in a sensitive and selective immunosensor for the detection of apolipoprotein-A1 [49].

A conventional electrochemical system for apolipoprotein-A detection was investigated by Liu et al. (2017). It consists of three electrodes, with the working electrode being made of glassy carbon modified with reduced graphene and multi-walled carbon nanotubes, the reference electrode made of Ag/AgCl, and the counter electrode made of platinum. On the working electrode surface, 100 μg/mL of human anti-Apo-A nanobody was added and incubated for 2 h. The functionalization of the working electrode was monitored by Cyclic Voltammetry and EIS measurements (Figure 2). Different concentrations of human Apo-A (0.5–1000 ng/mL) were analyzed using the Chronoamperometry technique and the detection limit was 0.2 ng/mL. The biosensor was tested for specificity with 1 μg/mL BSA, human Apo-E, human Apo-B, and 0.5 μg/mL human Apo-A, and the current response was higher only for human Apo-A [50].

### 4.3. Von Willebrand Disease

Von Willebrand factor (VWF) is a blood glycoprotein involved in hemostasis and thrombosis. Different CVDs, such as aortic stenosis, hypertrophic obstructive cardiomyopathy, or congenital structural diseases can generate high shear stress in the bloodstream, leading to the excessive cleavage of VWF multimers, known as acquired von Willebrand syndrome [51]. VWF is known to contribute to atherosclerosis and is associated with an increased risk of coronary heart disease and ischemic stroke [52]. In patients with pre-existing vascular disease, VWF is predictive of adverse cardiac events, including death.

A llama-derived nanobody (AU/VWFa-11) that recognizes an active von Willebrand factor was obtained, which showed good selectivity for the A1 VWF domain, as demonstrated by SPR analysis. In addition, they examined whether the AU/VWFa-11 nanobody could be used to detect an active VWF in patient plasma, as it was effective in detecting it in solutions, and the results were favorable [53].

Prediction of thrombocytopenia and bleeding risk in patients with von Willebrand disease type 2B was studied by performing the quantification of the von Willebrand factor from plasma samples in its GPIb-α binding conformation using the AU/VWFa-11 nanobody. The method used for this determination was immunosorbent assay in which microtiter wells were coated with the nanobody and were incubated with plasma samples diluted in phosphate-buffered saline and the amount of bonded VWF was determined with horseradish peroxidase-conjugated anti-VWF polyclonal antibodies [54].

Mutations in the A1 domain of the von Willebrand factor were detected using the same AU/VWFa-11 nanobody. The assay was performed on a family of 24 members, 11 of whom had a VWF factor due to a mutation of exon 28 of the VWF gene and 13 of whom served as controls. The nanobodies recognize the mutated monomers of VWF from the plasma of the affected family members [55].

## 5. Nanobodies for Cardiovascular Disease Therapy

Nanobodies are considered the primary candidates for new therapeutic strategies and drug delivery systems due to their ability to bind to epitopes that are not accessible to normal antibodies, through their long complementarity-determining regions (CDR) and their versatility. Using gene manipulation strategies can obtain nanobodies with identical or superior affinity and specificity to ordinary antibodies [56].

One of the cardiac diseases that affect cardiac functions and can be life-threatening is light chain amyloidosis (AL), which consists of the fibrillar deposition of misfolded immunoglobulin light chains (LCs). One of the most recent approaches for fighting this serious condition is based on targeting these immunoglobulin fragments by using nanobodies. The first experiments were carried out on a cardiotoxic light chain H3, and a series of analyses such as isothermal titration calorimetry, microscale thermophoresis, bio-layer interferometry, and multi-angle light scattering revealed that all the synthesized nanobodies bind to H3 and form stable complexes. As a result, of this interaction, H3 can no longer exhibit cardiotoxicity, as demonstrated in an experiment involving Caenorhabditis elegans [57].

Another very important target for cardiovascular medicine is the G-protein coupled receptors (GPCRs) and various nanobody-based strategies have been employed in order to study their structure, pharmacokinetics, and expression modulation [58]. By using a library of synthetic camelid antibody fragments, a series of nanobodies that can interact with angiotensin II type 1 receptor (AT1R) were discovered. The in vivo study on mice revealed that the nanobody antagonist has a similar antihypertension effect as compared to a known angiotensin receptor blocker, namely losartan [59]. Further studies showed that, for instance, clone NbAT110 helped determine the structural characteristics of the AT1R receptor and revealed how peptides bind and activate it, which until now, because of the size and flexibility of the peptides that bind to this receptor, was not very clear. Furthermore, some new information regarding receptor-peptide binding was revealed, such as a partial β-barrel at the N-terminal end of the receptor, a β-hairpin, and the N-terminus of the binding peptide [58].

Other scientists targeted sodium channels, known as Navs, which are involved in the rapid triggering of action potentials in both skeletal and cardiac muscles. It has been demonstrated that Nav mutations are the main cause of a series of genetic diseases, such as hypokalemic periodic paralysis, myotonia, and long-QT and Brugada syndromes. Although it has been proven that the development of highly specific nanobodies, targeting Navs can be very difficult due to their potential cross-reactivity. However, one team managed to obtain two nanobodies (Nb17 and Nb82) that appear to bind with nanomolar affinity to Nav 1.4 and Nav 1.5 isoforms, which belong to the skeletal muscle and, respectively, the cardiac muscle. The interaction between the two nanobodies and their specific targets in live cells was also demonstrated through a flow cytometry-based experiment, involving FRET and a two-hybrid screening technique. They demonstrated that FRET occurred for both cerulean-tagged Nb 17 and Nb82 in human embryonic kidney cells that express Nav1.5 channels [60].

Nanobodies have been generated to target the specific phosphorylation site of ryanodine receptor 2 (RyR2), in order to prevent the increased leakage of Ca^2+^ of sarcoplasmic reticulum (SR), which can determine cardiac dysfunction. RyR2-specific nanobodies were shown to inhibit the phosphorylation in vitro and further, for in vivo models (rat model). Investigations revealed that by mediating the expression of the AR185 protein, systolic and diastolic dysfunction has been improved, and the transfer of the AR185 gene, in affected cardiomyocytes, reduces SR calcium leaks. Laser scanning confocal microscopy was employed for the measurement of the SR Ca^2+^ levels in cardiomyocytes after incubation with Fluo-5N. These experiments showed that intracellular treatment using nanobodies can be a very efficient treatment tool for heart disease in rats; however, there is no proof yet that this strategy can be applied to human patients [61].

Modifications of the plasminogen activator inhibitor-1 (PAI-1) are often associated with cardiovascular disease. PAI-1 is a member of the serine protease inhibitor family, and its primary function is to inhibit urokinase and tissue-type plasminogen activators. Many different kinds of molecules, including antibodies and peptides, have been produced to serve as PAI-1 inhibitors, but their effectiveness is constrained by the molecule’s conformational plasticity. In recent years, the literature has also described nanobody libraries in addition to a series of PAI-1-interacting antibodies [62]. The Nb93 nanobody can bind to the reactive center loop (RCL) on the surface of PAI-1 and practically block the PA binding site of the molecule. According to a mutagenesis study, the epitope that the nanobody is targeting is positioned close to the RCL and includes the amino acids Glu242, Glu244, Glu350, and Lys243 [63].

Obesity is a very dangerous condition, usually associated with metabolic syndrome and a series of disorders such as cardiovascular diseases, type 2 diabetes, and hypotension. Although some treatments for this condition have been developed and approved, they were shown to have many adverse effects and for this reason, some of them were withdrawn. In an attempt to solve the lack of combat methods, scientists tried to analyze receptor GPR75 (G Protein Receptor), a new target for the treatment of obesity, using an intracellular nanobody. Using cryo-electron microscopy (Cryo-EM), they studied the GPR75-nanobody complex (NbH3) and were able to obtain information about the activation of the receptor, which is crucial information in the race for a new therapeutic agent anti-obesity [64].

The condition known as non-alcoholic fatty liver disease (NAFLD) is correlated with hypercholesterolemia and abnormal expression of a lipometabolic regulator, known as angiopoietin-like protein 3 (ANGPTL3). ANGPTL3 is mainly produced in the liver and can inhibit the enzyme activity of lipoprotein lipase and endothelial lipase. Recent findings reported the discovery of a VHH-Fc fusion protein that can bind ANGPTL3 with high affinity, which could be used as a therapy tool for treating patients with NAFLD. The obtained fusion protein, called C44-Fc, appears to help block the inhibition of the lipoprotein lipase by ANGPTL3. Tests were performed on hypercholesterolemic C57BL/6 mice, and after a dose of about 10 mg to 25 mg/kg of C44-Fc fusion protein, they had significantly lower levels of serum lipids, with LDL cholesterol levels decreasing by as much as 54.4% [65].

Studies show that heart failure is mainly caused by abnormalities in the Ca^2+^ signaling, which translates into modifications in the contraction and relaxation states. This condition arises from modifications in the activity of the sarco-endoplasmic reticulum Ca^2+^-ATPase (SARCA2a) and elevated levels of non-phosphorylated phospholamban, which is involved in its downregulation. For this reason, phospholamban is a target for new therapeutic approaches, and scientists have discovered VHH nanobodies that can bind, with very high specificity, to phospholamban in different conformational states and modulate its function. They used an adenovirus to transport this nanobody and demonstrated in a murine model that it can improve cardiomyocyte activity [66].

Nanobodies can also be used to help extracellular vesicles or any other type of drug carrier reach a desired target. One study used anti-EGFR nanobodies, of approximately 31 kDa, to guide red blood cells vesicles (RBCEVs), and their results revealed high affinity of the construct to EGFR-positive cell lines such as H358 and HCC827 (cancer lung cell lines) (Figure 3). The conjugation of the anti-EGFR nanobody to the RBCEVs was facilitated and optimized by using a linker peptide, designed to have glycine residues at the N-terminus and a binding site for sortase/ligase at the C terminus [67].

Thrombotic thrombocytopenic purpura (TTP) is a fulminant, rare disorder that implies the formation of blood clots in small vessels. This condition appears because of an acquired or congenital inability to process a very large protein called von Willebrand factor (VWF) by the protease ADAMTS13. In a healthy individual, the ultra-large VWF is cleaved by ADAMTS13, whereas in patients suffering from TTP, the inability of the protease to process these multimers, leads to excessive platelet aggregation and thrombi formation. Scientists had found a nanobody (anti-VWF Nanobody ALX-0681), that has the ability to impair VWF activity and prevent spontaneous platelet aggregation, in a baboon model [68].

Recently, the first nanobody-based (caplacizumab) treatment for TTP was approved by EMA and FDA. The ALX-0681 treatment is administrated subcutaneously and ALX-0081 by intravenous administration. The results obtained in phase III of the clinical trials showed that the treatment with ALX-0681 leads to faster disease resolution, reduced mortality, and fewer cases of recurrence [4].

More recently, anti-matrix metalloproteinase-2 nanobodies were developed to be used as highly selective inhibitors of this protease, which was shown to be involved in serious ailments such as cancer and cardiovascular disease. Marturano et al. (2019) showed that one nanobody (VHH-29) can inhibit collagen-induced platelet aggregation, whereas another nanobody (VHH-136) did not affect it. The inhibition effect of nanobody VHH-29 on MMP-2 was demonstrated to be quite selective because, for instance, the enzymatic activity of MMP-9 was only mildly affected. These findings are of particular importance because MMP-2 is a key factor for atherogenesis, plaque instability, and platelet activation, and so far, no other specific inhibitor has been found [69]. Another antiplatelet target is the collagen receptor glycoprotein VI, targeted with different nanobodies, and the results obtained from the western blot and whole blood thrombus formation evaluation suggest that Nb 2 nanobody blocks thrombus formation and platelet activation and is an excellent candidate for anti-thrombotic therapy [70].

## 6. Discussions

Among the advantages of nanobodies as opposed to antibodies, the most important ones are their ability to access hard-to-reach sites, bind to different regions of target molecules, offer high specificity and reach very low sensitivity, exhibit no secondary effects due to their fast elimination from the body, and of course the fact that they can be produced easily and with reduced costs. Due to the short blood half-life, researchers were able to use single-photon emission computed tomography imaging with ^99m^Tc-labeled nanobodies, 3 h after injection. Some of the most significant biophysical properties of nanobodies in relation to CVD applications in the field of diagnosis and therapy are presented in Table 1.

The development of nanobodies mainly takes place in mammalian cells, but due to the high effective costs and time consumption, applications in microorganisms, such as yeasts or Escherichia coli, are very attractive, especially for the pharmaceutical industry. Yeast cells are tools for various molecular nanobody applications and YSD is a widely used method for obtaining them in a quantitative and highly qualitative manner.

So far, many nanobody-based diagnostic tools have been developed, with demonstrated specificity to the desired target and excellent sensitivity of the nano and picomolar levels. However, although these techniques could be beneficial for early diagnosis and prevention of CVDs, they still require extensive work for the validation of in vivo models and often encounter ethical issues that make the process very slow. In some cases, advanced imaging techniques using nanobodies can encounter problems such as low binding efficiency [38,45] or low selectivity and discrimination from the background signal in the targeted area [40].

Whether they are used to activate or block a specific biomarker, as labeling molecules, or to help transport drugs to a specific target, nanobodies were successful in a series of applications using various techniques, and they appear to be versatile, as demonstrated by a great number of scientists. As previously demonstrated, they can even help determine the structural characteristics of target molecules or act as efficient inhibitors. Nanobody implication in the development of new and efficient CVD therapies has already been successfully applied in the case of patients suffering from Thrombotic thrombocytopenic purpura and the market approval of caplacizumab may be the start of a new trend for therapy management of patients with CVDs. However, due to the fact that nanobodies are not naturally produced in the human body, some questions arise about their safety as therapy tools, mainly because of their potential involvement in undesired immune responses. In order to avoid the potential negative effects, scientists have suggested surface modification of certain regions with human sequences, but this can lead to low solubility [73,74].

Advances in nanobody-based probes for molecular imaging could transform clinical medicine by uniting advances in the biological understanding of disease with progress in imaging technology. The ability to detect atheromas, which likely result in thrombotic problems in an economical, sensitive, and radiation-free manner, is realized by molecular imaging [75]. Additionally, studies have demonstrated that by coupling positron emission tomography (PET) with nanobodies targeting different receptors, information regarding the physiological status of cells affected by atherosclerosis can be obtained.

## 7. Conclusions

As demonstrated by a series of studies, nanobodies are becoming serious candidates for the development of new, accurate, sensitive, and cost-effective diagnostic devices due to their previously described advantages. Although most nanobody applications are in the early stages of development, the prevention and diagnosis of certain cardiovascular diseases such as atherosclerosis, coronary artery disease, thrombosis, and vascular diseases can be achieved by using the superior selectivity and specificity of nanobodies to target specific markers and receptors. In the field of imaging, they represent high-quality probes, that coupled with advanced imaging techniques, can offer valuable information on disease progression, specific interactions, and therapy efficacy.

Nanobodies can also serve for disease studies which can unravel the mechanisms behind it by understanding receptor functions, activation mechanisms, interactions, etc., and can act as inhibitors for certain disease-related factors or serve as “guides” for the transportation of drug-loaded nanocarriers to their desired target.

## Figures and Tables

**Figure 1 pharmaceuticals-16-00863-f001:**
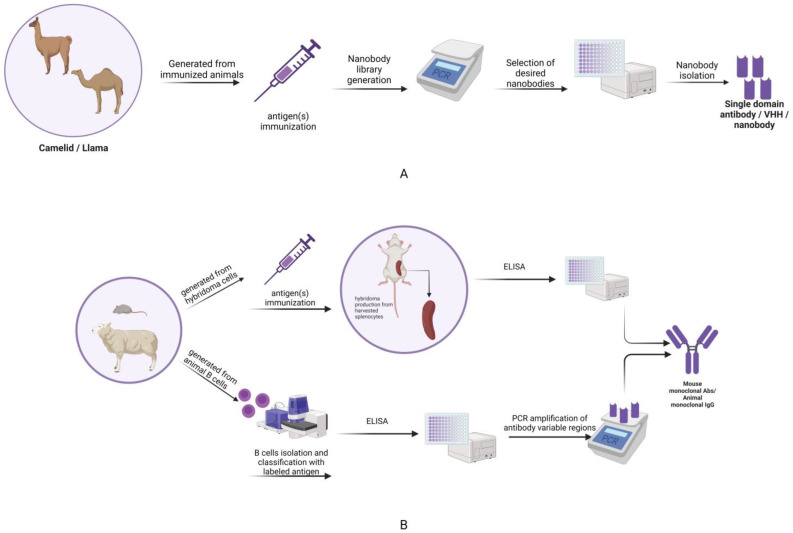
(**A**) Schematic representation of the development of nanobodies from previously immunized animals (camelids), including the generation of nanobody immune libraries and the further selection and isolation of the desired nanobodies that are going to be used through methods such as ELISA. (**B**) Schematic representation of the generation of antibodies by two highly acknowledged methods, one from previously immunized animals/mice and the other from animal B cells. This figure shows the steps required to obtain IgG/mouse monoclonal antibodies. The mouse monoclonal Abs are obtained by hybridoma technology and the animal monoclonal Abs are obtained by isolating and labeling the B cells with labeled antigen, followed by in vitro screening and then the PCR amplification of antibody variable regions in order to obtain the desired antibodies (adapted from Laustsen et al., 2021 [18] and created with BioRender.com, accessed on 5 April 2023).

**Figure 2 pharmaceuticals-16-00863-f002:**
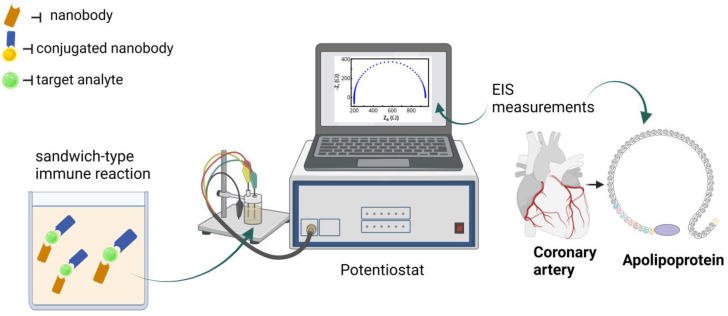
Electrochemical detection of apolipoprotein using nanobodies. Created with BioRender.com, accessed on 19 April 2023.

**Figure 3 pharmaceuticals-16-00863-f003:**
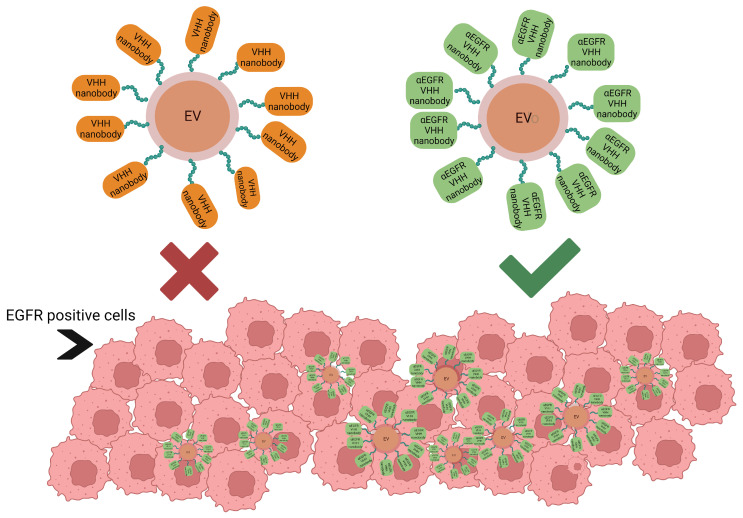
Extracellular vesicles, loaded with drugs, using peptide-coupled nonspecific nanobodies for the targeting of EGFR-positive cells versus anti-EGFR nanobodies. Created with BioRender.com, accessed on 24 February 2023.

**Table 1 pharmaceuticals-16-00863-t001:** Biophysical properties of nanobodies used in diagnosis and therapy of CVDs.

Biophysical Properties	Benefits	Drawbacks	Application
Small size	-can reach sites inaccessible to normal antibodies -rapidly eliminated from the blood due to their lower molecular weight -easily penetrates intercellular spaces and tissues	Rapid kidney elimination due to their molecular mass below threshold of glomerular filtration [4]	Advanced imaging for detecting atherosclerosis Therapies that target immunoglobulins, GPCRs, etc.
Strong antigen-binding affinity	-can bind different biomarkers and receptors	Poor design can lead to unspecific binding	Study of receptors structure pharmacokinetics and expression modulation Inhibition of CVD markers
Water solubility	-injectable solutions -preparation of bioconjugates	-	Imaging/Therapy Diagnostic methods: ELISA, immunosensors, electrochemical sensors
Extended CDR3 loops	-reversible conformation changes after thermal/chemical denaturation	Unwanted complexations may occur	Advanced imaging for detecting atherosclerosis
Single domain nature	More specific epitope binding, low cross-reactivity High affinity Increased stability	Difficulty in binding small antigens (haptens and peptides) [71]	Diagnostic–immunosensors and biosensors Targeting specific markers for therapeutic purpose
Low immunogenicity	-most probably will not induce undesired immune response	For tumor therapy, they still need to be humanized to ensure safety [72]	CVD therapies and in vivo diagnostic applications

## Data Availability

Not applicable.

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
