# Peer review of "Nanobodies as Diagnostic and Therapeutic Tools for Cardiovascular Diseases (CVDs)"

_pharmaceuticals, 2023, doi:10.3390/ph16060863_

Round 1

Reviewer 1 Report

There are several recent published papers on the use of VHH compounds as imaging and therapeutic agents already. To be fit for publication on an international scientific journal, this draft needs to be substantially reduced in size and to focus and elaborate on the themes described in the title.  They need to aim on the unique physico-chemical attributes of this relatively novel scaffold as they relate to its ongoing or potential applications in cardiovascular diseases.

The draft suffers from suboptimal syntax, requiring a better and more comprehensive description of the concepts described. The style of the narrative is frequently personal, with exaggerated use of adjective across the manuscript. 

Author Response

Dear esteemed reviewer,

We want to thank you for all the valuable suggestions, which in our opinion helped improve the manuscript and make it more accessible for the readers. Please find attached a pdf document containing all the modifications of the manuscript.

Kind regards,

The authors

Reviewer 2 Report

The diagnosis and treatment of cardiovascular disease (CVDs), even nowadays, remains a wishful scope, due to the increased number of people affected, the variety of CVDs, multiple factors that led to them, and inequalities in society regarding access to medical facilities. In recent years, camelid-derived single-domain antibodies, known as nanobodies gained much attention due to their superior properties compared with antibodies. Nanobodies are slowly produced in mammalian cells and relatively easy in prokaryotes, lower eukaryotes, and even plants. The model organism Saccharomyces cerevisiae is widely used for the development of nanobodies at the cell wall surface using the Yeast Display Surface method. Radiolabeled nanobodies are used in the non-invasive detection or screening of biomolecules involved in the progression of atherosclerosis, one of the most prevalent diseases that led finally to CVDs and to identify coronary heart disease markers. As nanobodies are considered the primary candidates for new therapeutic strategies, they are used to treat various CVDs or connected diseases, activate or block a specific biomarker, label molecules, or help transport drugs at a specific target.

The authors proposed a mini-review focused on the involvement of nanobodies in diagnosing and treating cardiovascular diseases.

The mini-review is interesting.

I have some minor comments:

1)     The abstract is not well written: the aim is at the end for example. Please better summarize the sections.

2)     Insert the key questions that the mini review must answer and a clear purpose.

3)     Insert the methods describing the search methodology and used criteria.

4)     Check the structure. The par “Nanobodies generated in yeast cells as a tool for molecular applications” for example is without the number.

5)     Discussion must be expanded with the limitations of your study and the limitations found in the references you analyzed.

6)     Please write the conclusions in a separated setcion

Author Response

(The authors gave the same response as above.)

Round 2

Reviewer 1 Report

This review is not acceptable for publication in its current, modified form.  Section 2 and 3 are of no critical relevance to the use of this novel scaffold for potential applications in cardiovascular diseases, and therefore should be significantly reduced in size.

The constant comparison with regular monoclonal antibodies appears biased since some of the examples described are VHH-Fc fusion constructs.  Clearly one of the major advantages of Nanobodies is, in many potential therapeutic applications, also a disadvantage (low exposure and fast Clearance). The quality of the diagrams is poor (not readable), and the organization of the contents in the original table is cramped, with mostly an unnecessary comparison (yeast vs. pichia vs. saccharomyces cerevisiae) for the intended audience.  This information belongs better in a commercial scientific catalog.  What is the relevance in their context of this manuscript?

The draft abuses the use of adjectives and a solid improvement in the overall syntax across the manuscript is needed.  The revised abstract is a perfect successful  example of this, and it should be extended to the entire manuscript.

The draft abuses the use of adjectives and a solid improvement in the overall syntax across the manuscript is needed in a similar fashion as it was done with the original abstract.

Author Response

Dear esteemed reviewer,

Thank you for all your valuable suggestions. The manuscript has been revised according to the recommendations. The modifications can be easily followed due to track changes.

Kind regards,

The authors
